# Mitochondria in Neuroprotection by Phytochemicals: Bioactive Polyphenols Modulate Mitochondrial Apoptosis System, Function and Structure

**DOI:** 10.3390/ijms20102451

**Published:** 2019-05-17

**Authors:** Makoto Naoi, Yuqiu Wu, Masayo Shamoto-Nagai, Wakako Maruyama

**Affiliations:** Department of Health and Nutrition, Faculty of Psychological and Physical Science, Aichi Gakuin University, Nisshin, Aichi 320-0195, Japan; wyuqui@gmail.com (Y.W.); nagaim@dpc.agu.ac.jp (M.S.-N.); maruyama@dpc.agu.ac.jp (W.M.)

**Keywords:** phytochemicals, mitochondria, oxidative stress, apoptosis, neuroprotection, anticancer, mitogenesis, fission/fusion, mitophagy

## Abstract

In aging and neurodegenerative diseases, loss of distinct type of neurons characterizes disease-specific pathological and clinical features, and mitochondria play a pivotal role in neuronal survival and death. Mitochondria are now considered as the organelle to modulate cellular signal pathways and functions, not only to produce energy and reactive oxygen species. Oxidative stress, deficit of neurotrophic factors, and multiple other factors impair mitochondrial function and induce cell death. Multi-functional plant polyphenols, major groups of phytochemicals, are proposed as one of most promising mitochondria-targeting medicine to preserve the activity and structure of mitochondria and neurons. Polyphenols can scavenge reactive oxygen and nitrogen species and activate redox-responsible transcription factors to regulate expression of genes, coding antioxidants, anti-apoptotic Bcl-2 protein family, and pro-survival neurotrophic factors. In mitochondria, polyphenols can directly regulate the mitochondrial apoptosis system either in preventing or promoting way. Polyphenols also modulate mitochondrial biogenesis, dynamics (fission and fusion), and autophagic degradation to keep the quality and number. This review presents the role of polyphenols in regulation of mitochondrial redox state, death signal system, and homeostasis. The dualistic redox properties of polyphenols are associated with controversial regulation of mitochondrial apoptosis system involved in the neuroprotective and anti-carcinogenic functions. Mitochondria-targeted phytochemical derivatives were synthesized based on the phenolic structure to develop a novel series of neuroprotective and anticancer compounds, which promote the bioavailability and effectiveness. Phytochemicals have shown the multiple beneficial effects in mitochondria, but further investigation is required for the clinical application.

## 1. Introduction

More than 1.5 billion years ago, mitochondria originate from bacterial endosymbions within some ancestral type of eukaryotic cells containing the nucleus, cytoskeleton, and endomembrane system, but lacking mitochondria [1]. Analysis of mitochondrial genes and genomic organization indicates that mitochondrial genes are derived from a α-protebacterium-like ancestor. The rapid evolution established mitochondria as compartmentalized factories of bioenergy and biosynthesis in the eukaryotic cells: cellular energy production, calcium (Ca^2+^) and iron homeostasis, cell differentiation, apoptosis, autophagy, sirtuin induction, and synthesis of protein, lipids, heme, and iron-sulfur clusters. Mitochondria produce also reactive oxygen species (ROS) through oxidative phosphorylation in the electron transport chain (ETC). Mitochondria are implicated in regulation of cellular redox potency, which is important for normal physiological processes [2], the deregulation of which is associated with the pathogenesis of aging, neurodegenerative diseases, such as Parkinson’s and Alzheimer’s disease (PD, AD), cardiovascular diseases, inflammation, and metabolic disorders.

In PD and AD, apoptosis is often detected in the brain [3]. Apoptosis is classified into mitochondria-activated intrinsic and receptor-mediated extrinsic apoptosis. Oxidative stress, ATP depletion, neurotoxins, excitotoxicity, and deficit of neurotrophic factors (NTFs) activate mitochondrial apoptosis cascade [4]. Apoptosis is initiated with mitochondrial membrane permeability transition (MPT), a sudden increase in membrane permeability, followed by opening of the non-specific mega-channel called the mitochondrial permeability transition pore (mPTP), and release of apoptosis-executing molecules, such as cytochrome c (Cytc), second mitochondria-derived activator of caspases (Smac)/Diablo, apoptosis-inducing factor (AIF), and endonuclease G, from the matrix to the cytoplasm. Apoptosis proceeds sequentially by activation of caspases and cleavage of nuclear DNA, leading to cell death with typical morphological feature. The outer membrane transporter protein (18 kDa) (TSPO) mediates the MPT. The tryptophan-rich sensory protein (TspO) was found to regulate the transition between photosynthesis and respiration in the carotenoid gene cluster of the photosynthetic bacterium *Rhodabacter*. The function of the TSPO and TspO is conserved through revolution, suggesting that TSPO may be developed from TspO the ancient bacterial receptor/stress sensor to play roles in the outer mitochondrial membrane (OMM) of eukaryotes [5,6]. These results suggest that mitochondria have their own death system to decide the fate of host cells and function as a foe within.

Mitochondria and their apoptosis system are targets of neuroprotective therapy to prevent neuronal loss and enhance existing cellular function [7]. Mitochondriotropic neuroprotective compounds include antioxidants (vitamin E), mitochondria-targeted antioxidants (MitoQ), mitochondria-activating compounds (creatine, coenzyme Q10), and the beneficial effects have been proven in cellular and animal models [8]. However, other cell organelles regulate mitochondrial functions by ROS/reactive nitrogen species (RNS), signaling pathways, and the discharge and uptake of Ca^2+^, suggesting that neuroprotective therapy should object also these multi factors. A number of neuroprotective compounds have been reported, such as inhibitors of type B monoamine oxidase (MAO-B), selegiline [(-)deprenyl, (2*R*)-*N*-methyl-1-phenyl-*N*-prop-2-ynyl- propan-2-amine] and rasagiline [(*R*)-*N*-2-prop-2-ynyl-2,3-dihydro-1*H*-indenamime], and molecular mechanisms of neuroprotection have been presented [9,10,11].

Epidemiological studies show that diet and bioactive food factors, to some extent, can ameliorate decline in cognitive, motor, sensory, and emotional function in aging, PD, AD, and depression. Phytochemicals, especially dietary polyphenols, have claimed to be responsible for health beneficial effects [12,13,14]. Antioxidant activity has been considered as the major neuroprotective function of polyphenols, but their concentration is too low to scavenge free radicals in the brain [15]. Novel antioxidant-independent neuroprotective functions have been confirmed: direct regulation of the MPT, modulation of cellular signal pathways, and induction of genes coding anti-apoptotic Bcl-2 protein family and NTFs, especially brain-derived and glial cell line-derived neurotrophic factor (BDNF, GDNF) [16,17,18].

This paper presents molecular mechanisms underlying anti-apoptotic and neuroprotective functions of dietary polyphenols in mitochondria. Polyphenols suppress or activate mitochondrial death system in post-mitotic neuronal cells and fast-growing cancer cells. Polyphenols also affect mitochondrial function and structure by modulating the biosynthesis (mitogenesis), dynamics (fission, fusion), transport, and autophagic cleavage of damaged mitochondria (mitophagy). Involvement of the redox and amphipathic properties of polyphenols in the dualistic functions is discussed. Synthesis of mitochondriotropic phytochemical derivatives is presented to develop novel series of neuroprotective and anticancer agents.

## 2. Oxidative Stress and Mitochondrial Dysfunction are the Major Pathogenic Factors in Neurodegeneration

### 2.1. Production of Reactive Oxidative Species and Mitochondrial Dysfunction

In neurodegenerative diseases, oxidative stress and mitochondrial dysfunction fall in a vicious cycle to cell death. In the ETC, complex I (NADH dehydrogenase-ubiquinone oxidoreductase) and III (ubiquinone-cytochrome c oxidoreductase) leak electrons to oxygen and produce superoxide, which is further transformed into hydrogen peroxide by superoxide dismutase (SOD) and peroxynitrite by reaction with nitric oxide. ROS and RNS produced in mitochondria are excreted into the cytoplasm by voltage-dependent anion channel (VDAC), and oxidize and modify protein, lipid, and DNA [19]. Dysfunction of complex II (succinate dehydrogenase-ubiquinone oxidoreductase) of the ETC also produces ROS conditionally. Components of tricarboxylic acid (TCA) cycle, α-keto-dehydrogenase and pyruvate dehydrogenase, generate superoxide in the matrix. In addition, in neuronal and glial cells, MAO is localized at the OMM, oxidizes monoamine neurotransmitters, and produces hydrogen peroxide. ROS generated in mitochondria act as signals controlling cell function under physiological condition, but the excessive and sustained increase induces cell death in aging and related disorders [2,20].

Defects of the ETC complexes and increased oxidative damage are consistent features of parkinsonian brain. 1-Methyl-4-phenyl-1,2,3,6-tetrahydropyridine (MPTP) caused parkinsonian syndrome in human, and 1-methyl-4-phenyl-pyridinium ion, produced from MPTP by MAO-B-dependent oxidation, inhibited the complex I activity [21]. The selective loss of dopaminergic neurons was detected in the substantia nigra of idiopathic PD and MPTP-Parkinsonism, and mitochondrial dysfunction was proposed as one of the major pathogenic factors. Lower complex I activity and selective decrease in a number of subunits of complex I were detected in mitochondria isolated from the parkinsonian frontal cortex, platelets, lymphocytes, and skeletal muscle [22]. In patients with AD, significant reduction in complex III and IV (cytochrome c oxidase) activity was reported in the hippocampus and platelets, whereas complex V (F_1_-F_0_-ATP synthase) activity did not change [23]. Oxidative stress is accepted as the primary pathogenic factor in neurodegeneration, but it has not been established whether mitochondrial dysfunction is the cause or consequence of disorders [24].

Complex I dysfunction increases ROS/RNS and inflammatory cytokines, and damages mitochondrial DNA (mtDNA), and induces the mutation. High leveled deletion of mtDNA encoding 13 protein subunits of complex I, III, IV, and V was found in neurons of the substantia nigra in aging, and the deletion was still higher in patients with early PD and incidental Lewy body disease [25]. Point mutation in 12SrRNA (T1095C) was detected in a pedigree with Parkinsonism with deafness and neuropathy [26] and mutation in G11778A in familiar parkinsonian patients with Leber’s optic atrophy [27]. Polymorphisms in mtDNA, such as A4336G [28] and excess of non-synonymous mutations in mtDNA-coded complex I [29], were reported to contribute to the pathogenesis of PD and AD. On the other hand, activated microRNA miR-144-3p upregulated expression of genes involved in mitochondrial function, including peroxisome proliferator-activated receptor γ (PPARγ) cofactor-1 (PGC-1α, nuclear respiratory factor-1 (Nrf1) and mitochondrial transcription factor (TFAM) [30].

Another typical pathological feature in AD and PD is the accumulation of disease-specific protein aggregates: α-synuclein (αSyn) in PD, and amyloid beta β(Aβ) and hyperphosphorylated tau in AD. The oligomers and fibrils, intermediates of the amyloidogenesis, inhibit mitochondrial activity and exhibit potent cytotoxicity. αSyn is localized mainly in the cytoplasm, but also in mitochondria. Binding to membrane lipid transforms αSyn into an amphipathic α-helical structure leading to fibril and oligomer formation. αSyn is recognized by translocase of outer membrane (TOM) and brought to the inner mitochondrial membrane (IMM) translocase (IOM) and finally into the matrix. In the parkinsonian brain, dopamine-modified αSyn-TOM (TOM40, TOM20) complexes were detected, which inhibited mitochondrial protein import, increased oxidative stress and impaired complex I activity [31,32]. The TOM was reported to transport Aβ into the mitochondrial cristae and TOM-Aβ complex was identified in human cortical brain biopsies [33]. Aβ and αSyn interacted with VDAC, cyclophilin-D (CypD), and other mPTP components and triggered the formation of the mPTP and apoptosis [34,35,36,37,38]. In PD and AD, oxidative stress and mitochondrial dysfunction are associated with neurodegeneration, and the oligomers and fibrils of PD- or AD-specific amyloids are accumulated in mitochondria and accelerate the dysfunction. Neuroprotective therapy primarily targets to break the vicious circle of oxidative stress and mitochondrial dysfunction.

### 2.2. Structure and Regulation of Mitochondrial Apoptosis System

This session reviews the structure and regulation of mitochondrial apoptosis system. Intrinsic apoptosis is detected in the brain from patients with neurodegenerative diseases, ischemic/perfusion injury, and toxic syndromes. The mPTP formation is the point of no return in apoptosis. However, the exact structure and molecular mechanism of the mPTP formation are still under debate, and the mPTP is now considered to assemble only under specific conditions [39,40]. The major components of the mPTP are VDAC at the OMM, adenine nucleotide translocator (ANT) at the IMM, and CypD at the matrix. In addition, anti- and pro-apoptotic Bcl-2 protein family, TSPO, hexokinase (HK)-I and -II are associated with VDAC on the cytoplasmic face of the OMM, and creatine kinases (CK) at the intermembrane space. Glycogen synthase kinase-3β (GSK-3β) phosphorylates VDAC and inhibits its interaction with HK. Recently the F_1_-F_0_-ATP synthase (F-ATPase), which controls the electrochemical gradient generated across the IMM, has been reported to form the mPTP. Mitochondrial ATP synthase consists of a globular domain protruding into the mitochondrial matrix (F_1_ domain) and an IMM-embedded domain (F_0_ domain). Ca^2+^ binds to the F-ATPase at the catalytic site, changes the conformation, and F-ATPase is translocated from the matrix to the IMM, binds to CypD, and forms the mPTP [41,42]. Figure 1 presents the schematic structure of the mPTP.

Under physiological conditions, the IMM is impermeable to most irons and solutes, whereas the OMM is quite permeable to ions, solutes, and even some small protein. Mild stimulus forms a transient and reversible pore at the IMM, which allows entry of water, solutes with molecular mass less than 980 Da, metabolites, and inorganic ions into the matrix, and transiently depolarizes mitochondrial membrane potential (ΔΨm) (Figure 2A,B). Cyclosporine A (CysA) inhibits CypD binding to ANT and prevents the pore formation at the IMM. More intense and prolonged stimulus irreversibly induces the mPTP formation, increases the IMM permeability to solutes with the molecular mass up to 1500 Da, Cytc and Ca^2+^, and causes expansion of the matrix and rupture of the OMM (Figure 2C), whereas anti-apoptotic Bcl-2 and Bcl-xL suppress this process by specific protein-protein interaction (Figure 2D,E). The mPTP is opened by polyanions and closed by polycations, suggesting the involvement of the IMM surface potential in the mPTP. In addition, the mPTP is formed at the interface between phospholipid polar head and surrounding aqueous milieu and associated with hydrophobic core of the IMM. These ionic and amphipathic properties of mitochondrial membrane are implicated in the selective affinity of mitochondria-targeting compounds to the mPTP. Even though numerous results were reported, the detailed nature and regulation mechanism of mitochondrial death systems remain elusive. To date, CypD and Bcl-2 protein are considered as main regulators of the mPTP formation.

## 3. Polyphenols Modulate Mitochondrial Function and Exhibit Neuroprotection

Epidemiological studies presented neuroprotective effects of the natural food products in aging, neurodegeneration, and depression. The risk of dementia was significantly inverse with flavonoid intake [43]. Polyphenols improve mitochondrial functions, especially the ETC activity, modulate the redox state and inhibit the apoptosis system. Mitochondria-targeting polyphenols have been devised to increase the selective accumulation of bioactive compounds in mitochondria [44,45,46]. Cellular mechanisms behind neuroprotection by polyphenols have been clarified to some extent in animal and cellular models of AD and PD, but clinical trials of intervention with individual phytochemicals could not present consistent results [47,48].

### 3.1. Structure and Anti-Oxidant Function of Polyphenols

Polyphenols scavenge free radicals, chelate metals, increase the activity and expression of antioxidant enzymes, and suppress those of ROS-producing enzymes, to function as neuroprotective antioxidants. Flavonoids are the most common bioactive polyphenols and more than 6000 types have been identified and they are characterized by a structure consisting of two benzene rings, (A and C rings) linked by a heterocyclic pyran or pyrone ring (B ring), each bearing at least one hydroxyl group, and connected with a three-carbon bridge (Figure 3). Flavonoids are divided into subgroups based on the degree of oxidation of the C ring, the hydroxylation pattern of the rings, and the substitution of position 3 in the C ring. The main six subgroups are (1) flavonols (e.g., kaempferol, quercetin, myricetin, rutin) found mainly in broccoli, onion, kale, and fruit peels, (2) flavanols [epicatechin (EC), epicatechin-3-gallate (ECG), epigallocatechin-3-gallate (EGCG)] in green tea, red wine and chocolate, (3) flavones (apigenin, luteolin) in apple skins, parsley, and celery (4) isoflavones (daidzen, genistein) in soybeans and legumes, (5) flavanones (hesperetin, raringenin) in citrus fruit and tomatoes, and (6) anthocyanins (cyanidin, delphidin, malvidin) in red wine, cherries, grapes, and berry fruits. The chemical structures of major flavonoids are shown in Table 1. Several non-flavonoid polyphenols, curcumin [1,7-*bis*(4-hydroxy-3-methoxy-phenyl)-1,6- heptadiene-3,5-dione], phenolic acids, stilbenes [resveratrol (3,5,4′-trihydroxy-*trans*-stilbene)], astaxanthin (3,3′-dihydroxy-β,β-carotine-4,4′-dione), and sesame lignans (sesamin, sesaminol, sesamolin) are also neuroprotective antioxidants, and their structures are shown in Figure 4.

Several flavonoids (EC, EGCG, naringenin, hesperetin, anthocyanins) can cross the blood-brain-barrier (BBB) in vivo and the BBB model composed of ECV304 cells grown on filters above C6 glioma cells [49,50]. In flavonoids, *ortho*-dihydroxyl substitution in the B ring and the presence of 2,3-unsaturation and a 4-carbonyl in the C ring increase stability of the phenoxyl radicals and electron delocalization [51]. The radical scavenging activity of major tea catechin is in order of EGCG > ECG > EC. To chelate metals (iron, copper ions,) a 5- or 3-hydroxy substitution in the C ring is additionally required [52]. Chelating Cu^2+^ and Fe^2+^ prevents hydroxyl radical generation from hydrogen peroxide by the Fenton reaction. Flavonoids protect neurons by the antioxidant activity, modulation of signal cascades and induction of gene expression. Substitution of 3-hydroxy group in the C ring and 5- and 7-hydroxy groups in the A ring is associated with neuroprotective activity. Flavonoids increase ROS-removing enzymes (SOD, catalase, glutathione reductase) via activation of Keap1 (Kelch ECH-associated protein 1)/Nrf2/ARE (antioxidant responsive element) signaling pathway. Polyphenols (apigenin, catechin, kaempferol, luteolin, quercetin, curcumin) inhibit ROS-producing XO (xanthine oxidase), MAO, and NADPH oxidase (NOX) [53,54,55].

Curcumin is the yellow pigment present in *Curcuma longa* and it possesses antioxidant, anti-apoptotic, anti-inflammatory, anticancer, and anti-diabetic properties. [56]. It is composed of an aliphatic unsaturated heptene linker with two aromatic rings attached at the both ends. Curcumin is a potent hydrogen-atom donor, and the phenolic hydroxyl, β-diketo, and methylene group are required for the antioxidant activity. Lipophilic curcumin can scavenge hydrogen peroxide, hydroxyl radical, and peroxynitrite and prevent lipid peroxidation in vivo and in vitro [57]. Curcumin reduces ferric ion (Fe^3+^) and chelates ferrous ion (Fe^2+^).

Phenolic acids are classified into benzoic acid derivatives (gallic, vanillic, protocatechuic acid) and cinnamic acid derivatives (*p*-coumaric, caffeic, ferulic acid). The chemical structures are shown in Table 2. It is a major constituent of orange, tomato, carrot, and sweet corn. They have electron-donating 3-methoxy and 4-hydroxy groups on the benzene ring, and can scavenge hydroxyl and superoxide radical, peroxynitrite, and terminate radical chain reactions. The carboxylic acid binds to lipid bilayer and prevents lipid peroxidation with an adjacent unsaturated carbon–carbon double bond [58]. Phenolic acid and the ethyl ester derivatives can upregulate protective genes, such as heme oxygenase-1 (HO-1), heat shock protein 70, extracellular signal-related kinase 1/2 (ERK1/2). They are proposed as therapeutic and preventing agents in neurodegenerative disorders, cancer, cardiovascular diseases, diabetes, and skin diseases.

Resveratrol is a stilbene found in the skin of red grapes, raspberries, blueberries, red wine, and Japanese knotweed, and has multiple biological and pharmacological properties: antioxidant, antimicrobial, anti-inflammatory, neuroprotective, cardio-protective, anti-aging, and anti-tumor activities. It can scavenge hydroxyl radicals in vitro with its hydroxyl groups, and 4′-hydroxyl group is the most reactive. However, the antioxidant potency in vivo is quite low, and the antioxidant activity may be mainly due to activation of Nrf2/ARE pathway and induction of HO-1 and SOD [59].

Astaxanthin occurs in algae, krill, trout, and salmon and can cross the BBB and have antioxidant, anti-inflammatory, anti-tumor, and anti-apoptosis activities by upregulation of protective gene expression, such as Bcl-2. It is the polyene chain, and the nonpolar middle segments are composed of carbon–carbon double bonds present in transmembrane orientation, whereas each of the cyclic end rings can be esterified at the hydroxyl and keto moieties and increase antioxidant potency. These two unique characters provide astaxanthin to scavenge superoxide and hydrogen peroxide at the surface and suppress lipid peroxidation inside the membrane [60].

Sesame lignans are found in sesame (*Seamum indicum* L.) seeds with anti-oxidative, anti-hyperlipidemic, and anti-hypertensive activities, and major lignans are sesamin, sesaminol, sesamolin, and sesamolinol. They scavenge superoxide and reduce lipid peroxidation in vitro, and most active lignan against hydrogen peroxide in vitro was sesamolin, followed by sesaminol and sesamin [61]. But, in vivo radical scavenging activity of sesamin and sesamolin was very weak [62], and the in vivo antioxidant function was ascribed to the upregulation of antioxidant enzymes (catalase, SOD, HO-1) and downregulation of oxidative enzymes (NOX-2, cyclooxygenase 2) [63]. Antioxidant activities are essential for bioactivity of polyphenols, but the in vitro results cannot be always reproduced in vivo, because of the limited bioavailability and intense metabolism. Antioxidant activity observed in vivo is mainly ascribed to enhance gene expression of antioxidants.

### 3.2. Polyphenols Prevent Apoptosis by Direct Regulation of Apoptosis System in Mitochondria

Polyphenols protect neurons in cellular and animal models, and quercetin, hesperidin, curcumin, and resveratrol are reported to have the most potent anti-apoptotic activity. Curcumin prevented mitochondrial apoptosis induced by ischemia/reperfusion in rats [64], and black tea theaflavin prevented apoptosis and the behavioral abnormalities in MPTP-induced C57BL/6 mouse model of PD [65]. Resveratrol and quercetin protected PC12 cells from MPP^+^-induced apoptosis [66]. Morin and mangiferin (*C*-glycosyl-1,2,6,7-tetrahydroxyxanthen-9-one) protected primary cultured rat brain neurons from glutamate-induced apoptosis [67]. Polyphenols prevented the MTP, ΔΨm decline, the mPTP opening and caspase activation, and prevented apoptosis. Astaxanthin, gallic acid, hesperidin, myricetin, rosmarinic acid, black tea theaflavin, and quercetin and pinocembrin [(2*S*)-5,7-dihdroxy-2-phenyl-2,3- dihydrochromen-4-one] and their methylated derivatives inhibited the mPTP opening induced by inorganic phosphate, Aβ42 peptide, oxygen-glucose-deprivation, ischemia/reperfusion, and neurotoxins (MPTP, homocystein) [68,69,70,71]. Anti-apoptotic functions have been considered through upregulation of Bcl-2 and Bcl-cL, downregulation of Bax and Bak, regulation of mitochondrial function, redox state, and Ca^2+^ homeostasis.

The effects of polyphenols on the mPTP have been determined indirectly by detection of increased levels of Ca^2+^ and Cytc and activated caspases in the cytoplasm. To investigate direct regulation of the mPTP by polyphenols, a cellular model of the MPT was prepared in dopaminergic neuroblastoma SH-SY5Y cells using a TSPO ligand PK11195 [1-(2-chlorophenyl)-*N*-methyl-*N*- (1-methylpropyl)-3-isoquinoline carboxamide]. PK11195 induced the mPTP opening and apoptosis, which Bcl-2 overexpression and pretreatment with CysA or rasagiline could completely prevented [9]. The step-wise formation of pore at the IMM and OMM was detected by measurement of superoxide and Ca^2+^ efflux from mitochondria into the cytoplasm, respectively [11,16,72]. Immediately after PK11195 addition, superoxide increased in a single peak with 3–5 s duration, whereas Ca^2+^ signal increased gradually and reached to plateaus at 2 min (Figure 2D,E). Bursts of superoxide production in the ETC causes rapid superoxide signal called “superoxide flashes”. CysA pretreatment could suppress superoxide flash completely, and Bcl-2 overexpression inhibited superoxide flashes and Ca^2+^ efflux completely. Ca^2+^ efflux was positively correlated with cell death, suggesting that the Bcl-2-regulated pore formation was the pivotal step to induce apoptosis. The effects of polyphenols on the MPT were investigated [16]. Pretreatment of astaxanthin prevented PK11195-induced superoxide flash and Ca^2+^ efflux, and also cell death (Figure 5A,B). The structure–activity relationship was investigated with ferulic acid derivatives, which share the same phenolic structure to scavenge ROS/RNS and have the same redox potency, but different amphiphilic side chain (Figure 5C). Ferulic acid and the aldehyde and alcohol derivative (coniferyl aldehyde and alcohol) suppressed Ca^2+^ efflux and cell death induced by PK11195, whereas the hydrophobic ethyl ester synergistically increased Ca^2+^ efflux and PK11195 cytotoxicity (Figure 5D). Among sesame lignans, hydrophobic sesamin increased superoxide flash and Ca^2+^ efflux in control and further increased those in PK11195-treatd cells. Hydrophilic sesaminol and sesamolin suppressed superoxide flash and Ca^2+^ efflux in PK11195-treated cells with the same potency regardless of the different antioxidant activity. The lipophilic derivatives of caffeic and ferulic acid induce the MPT and apoptosis in human breast cancer cells [73]. These results suggest that hydrophobicity of these polyphenols might be associated with apoptosis induction.

Polyphenols prevented the mPTP opening by interaction with the mPTP components (Figure 1). Curcumin interacted with amino acid residues in the helical *N*-terminal of VDAC, changed the conformation, and inhibited the mPTP opening [74]. Resveratrol promoted deacetylation of VDAC1 via Sirt1, inhibited the interaction with Bax, increased that to Bcl-2, and prevented the ΔΨm collapse and Cytc release [75]. Resveratrol and hesperidin activated Akt/GSK-3β pathway, inhibited VDAC1 phosphorylation, promoted dissociation from the complex composed with Bax, induced binding to HK-II, prevented the MPT and apoptosis caused by anorexia/reoxygenation in cardiomyocytes [76], and prevented Aβ-induced apoptosis in PC12 cells [69]. Biapigenin (4′,4′’’,5,5′’,7,7′’-hexahydroxy-3,8′-biflavone, found in cereals) activated ANT, enhanced Ca^2+^ efflux from mitochondria, reduced Ca^2+^ burden, modulated the mPTP, and protected cells against excitotoxicity [77]. Polyphenols affect the mPTP opening by direct interaction with the mPTP components, or intervention in posttranslational modification of the mPTP protein. However, the whole process of apoptosis is still not fully clarified. Further studies are required for understanding of the detailed mechanism, and our developed cell model [9,11,16,72] might be applicable for the future study.

### 3.3. Polyphenols Enhance Mitochondrial Biogenesis

Mitochondria are dynamic organelle responding to cellular conditions, and the biogenesis (mitogenesis), dynamics (fission and fusion), and autophagy (mitophagy) are important for proper mitochondrial function, distribution, structure, and movement. Mitogenesis is a process by which new mitochondria are produced by growth and division from existing mitochondria. Caloric restriction, exercise, cell cycle, and hormones (growth, luteinizing, follicle stimulating hormone) stimulate mitochondrial biogenesis. Excess production of ROS/RNS within mitochondria and oxidative damage in protein components of the ETC, lipids, and mtDNA impair mitogenesis [78], which are implicated in the pathogenesis of PD, AD, cardiovascular diseases and diabetes type II, and neuroinflammation [79]. Mitogenesis involves synthesis of the IMM and OMM and mtDNA-encoded protein, synthesis and import of nucleus-encoded protein, and replication of mtDNA. PGC-1α coactivators regulate signal pathways associated with mitogenesis [80]. AMP-activated protein kinase (AMPK) is a major upstream regulator of PGC-1α and mitochondria biogenesis (Figure 6) Phosphorylation or deacetylation activates PGC-1α upregulates Nrf2 and TFAM, promotes the binding to promoter regions of nuclear genes encoding subunits of the five complexes in the ETC, and also replicates mtDNA. Ca^2+^/CaMK/CREB (Ca^2+^/calmodulin-dependent protein kinase/cAMP response element-binding protein) pathway are involved in mitogenesis.

Polyphenols promote mitochondrial biogenesis and minimize mitochondrial dysfunction [81]. Resveratrol inhibited cAMP phosphodiesterases, increased cAMP, and activated cAMP/CaMK/AMPA pathway to deacetylate and activate PGC-1α via NAD^+^/Sirt1 (silent information regulator of transcription 1) signaling [82,83]. Resveratrol also activated protein kinase C epsilon (PKCε) and AMPK, increased NAD^+^ levels, and stimulated mitochondrial function, biogenesis, and dynamics [84]. EGCG promoted mitochondrial biogenesis in cells from subjects with Down’s syndrome by Sirt1/PGC-1α pathway and upregulation of Nrf1 and TFAM and mtDNA content [85]. Flavones (baicalein, quercetin, wogonin), isoflavones (daidzen, genistein), curcumin, and hydroxytyrosol (3,4-dihydroxyphenylethanol, present in olives) enhanced mitochondrial biogenesis by increased expression of Sirt1/AMPA/PGC-1α, complex IV in the ETC, and TFAM in vivo and in vitro [86,87]. Oleuropein, the main polyphenol isolated from extra virgin olive oil, increased expression of mtDNA, PGC-1α, complex II and IV, regulated mitochondrial function, mitogenesis, and dynamics through mitofusion 1 (Mfn1) and dynamin related protein 1 (Drp1) and attenuated oxidative stress in the hypothalamic paraventricular nucleus of spontaneously hypertension rats [88].

In contrast to many results obtained in animal models, the effects of polyphenols on mitogenesis in humans have been scarcely reported. Quercetin supplement increased the mRNA levels of sirtuin 1, PGC-1α, cytochrome c oxidase, and creatine synthase in skeletal muscle after 2-week administration in young adult males [89]. EC-rich cocoa intake increased levels of Sirt 1, PGC-1α, TFAM, and complex I and V in skeletal muscle samples obtained by biopsy from patients with type 2 diabetes and heart failure [90], and EC-rich dark chocolate significantly increased AMPA and PGC-1α in samples from normal sedentary subjects [91]. Polyphenols can affect mitophagy directly or indirectly by activation of the upper stream signaling cascades to regulate transcriptional factors related to the biosynthesis. Increased biosynthesis of mitochondria has been confirmed to improve mitochondrial function and contribute neuroprotection by polyphenols in vivo and in vitro.

### 3.4. Polyphenols Influence Mitochondrial Fission and Fusion

Mitochondria are continually remodeled through two opposite processes, fission and fusion. Mitochondrial fusion is a process of two adjunct mitochondria to form more elongated mitochondria, and the dysfunction was reported as a risk factor for neuronal loss in PD, AD, and HD [92]. Mitochondrial dynamics are regulated by three dynamin-related guanosine triphosphatases; Mfn1 and 2 localized at the OMM, and optic atrophy gene 1 (Opa1) at the IMM [93]. Mutation of *Mfn2* and *Opa1* was reported to cause neuropathies in Charcot–Marie disease type A and autosomal dominant optic atrophy, respectively [94,95]. PGC-1α and -1β, estrogen-related receptor α, and the transcription factor myocyte enhancer factor-2 induce Mfn expression and regulate the OMM fusion. Insulin and NF-κB increase expression of Opa1, which is involved in IMM and OMM fusion and cristae remodeling. Loss of Opa1 was reported to disrupt mitochondrial cristae and induce spontaneous apoptosis.

Mitochondrial fission generates two mitochondria by division of a mitochondrion, and it is essential for cells to maintain adequate number of mitochondria. Drp1 exhibited specific fission activity on mitochondrial membranes, and mitochondrial fusion factor, mitochondrial dynamics protein of 49 and 51 kDa (MiD49 and MiD51) mediated the recruit from the cytoplasm to the OMM [96]. *Drp1* mutation caused fission deficit in mitochondria and peroxisomes in a lethal neonate case with microcephaly [97]. The activity and translocation of Drp1 to mitochondria are regulated by post-translational modification: phosphorylation, modification by small ubiquitin-like modifier, ubiquitination, and *S*-nitrosylation.

Curcumin promoted mitochondrial biogenesis and dynamics in vivo and in vitro [98]. Resveratrol pretreatment prevented decrease in Opa1, Mfn2, Drp1, and Fis1, increased mitochondrial biogenesis and fusion, mtDNA copy number and mitochondrial mass, and protected neuronal cells in rotenone-treated rats [99]. Resveratrol also protected hippocampal neurons in senescence-accelerated prone mouse (SAMP8) models [100]. Resveratrol also activated Sirt1, mitochondrial Sirt3, and Foxo3/PINK1 (PTEN-induced kinase 1)/Parkin pathway, maintained the balance between fusion and fission, prevented mitophagy, and protected cells [101]. Anthocyanins stabilized the fusion/fission processes and protected neuronal cells against cytotoxicity of rotenone and amyloid precursor protein (APP)swe mutation [102], and attenuated Opa1 cleavage and prevented apoptosis in cerebellar granule neurons [103]. Allicin (diallyl thiosulfinate), the main biological compounds derived from garlic, ameliorated Drp-1 upregulation and Opa1 downregulation in 6-hydroxydopamine (6-OHDA)-treated PC12 cells, and prevented fragmentation of mitochondrial network and apoptosis [104]. Mitochondrial dynamics are essentially associated with maintenance of cell function and survival. Polyphenols influence the signal pathways and post-translational modification of related factors, induce the expression of factors involved in the dynamics and protect cells.

### 3.5. Polyphenols Affect Mitophagy and Control Mitochondrial Quality

Mitochondria-targeted autophagy (mitophagy) cleaves damaged mitochondria, controls mitochondrial quality, and promotes cell survival in aging and neurodegenerative diseases. Autophagy degrades unwanted cytoplasmic aggregates and dysfunctional organelles in a lysosome-dependent way [105,106]. During specified developmental stages and starvation, even healthy mitochondria are subjected to mitophagy, and sequestered proteins, carbohydrates, and lipids are degraded to compensate for nutrient deprivation [107]. Mitophagy processes in sequential stages, fission, priming, and engulfment [108]. Starvation activates the serine/threonine protein kinase mammalian target of rapamycin (mTOR) and initiates autophagy, which is modulated by intracellular Ca^2+^ and inositol 3-phosphate (IP_3_) levels. mTOR kinase activates or represses autophagy through regulating transcription factors including p53, STAT3, and NF-κB, Phosphorylated AKT enhances vesicle nucleation to form complex with Bcl-2-interacting protein (Beclin 1), Vps34, autophagy-related gene 14 (Atg14), and molecules in Beclin1-regulated autophagy protein-1 and Bif-1/endophilin B. Conjugation of Atg12 to Atg5 mediates vesicle elongation and promotes the binding of microtuble-associated protein 1 light chain 3 (LC3) to phosphatidylethanolamine.

Selective autophagy receptors bind to the target to recruit autophagic machinery [109]. The hypoxia-inducible proteins, Bcl2-L-13 and Nix (Nip3-like protein X), were identified as mammalian mitophagy receptors. Bcl2-L-13 and Nix are localized at the OMM and interact with identical LC3/GABARAP (GABA receptor-associated protein) at the LC3-interacting region. Nix recruits GABARAP to damaged mitochondria to initiate mitophagy. This pathway is associated with mitophagy under conditions demanding high ATP supply and also during physiological processes, such as reticulocyte maturation and erythrocyte development. Nix binds and recruits the ULK1 (unc-51-like autophagy activating kinase 1) complex to induce mitophagy [110].

PINK1 and Parkin, whose mutations and loss of function are identified in autosomal recessive early-onset PD, are implicated in mitophagy. Mitochondrial dysfunction induces ∆Ψm collapse, PINK1 accumulation at the OMM, and ubiquitin phosphorylation, and activates Parkin the E3 ubiquitin ligase. Parkin polyubiqutinates mitochondrial proteins, induces their association with the ubiquitin-binding domains of autophagy receptors, and forms the autophagosomes [111]. Nix could restore mitophagy and mitochondrial function in cells derived from PINK1/Parkin-related PD patients [112]. Carnosic acid (a diterpene compound in rosemary) protected SH-SY5Y cells against toxicity of 6-OHDA by activating the PINK1/Parkin-mediated mitophagy [113].

Modification of autophagy by polyphenols has been proposed as a promising therapeutic strategy for neurodegenerative diseases and cancer [114,115]. Polyphenols modulate autophagy by canonical (Beclin-1 dependent) and non-canonical (Beclin-1 independent) signal pathways for neuroprotective or anticancer therapy [116]. In the canonical pathway, Beclin-1 and the regulatory subunits Vps34 and Vps15 form a protein complex, and phosphatidylinostol-3-kinase (PI3K) induces autophagosomal nucleation and autophagy. Curcumin downregulated PI3K/Akt/mTOR and mTOR/p70S6K signal pathways and activated autophagy and showed neuroprotection in APP/presenilin 1 double transgenic mice [117] and in A53T αSyn cellular model of PD [118]. Curcumin restored reduction of autophagy markers LC3I/II by paraquat and protected SH-SY5Y cells [119]. Oleuropein aglycone (the methyl ester of 3,4-dihydro-2*H*-pyran-5-carboxylic acid) activated AMPK, phosphorylated ULK1 at Ser555, inhibited mTOR pathway, and induced autophagy, and ameliorated cognition deficits and Aβ plaque pathology in TgCRND mice model overexpressing the Swedish and Indiana Aβ mutations [120,121,122]. Loganin (7-hydroxy-6- desoxyvenalin) a compound derived from several plant species downregulated LC3-II and Drp1 expression and autophagy, and protected neurons in MPTP-induced mouse model of PD [123]. Kaempferol increased LC3-II, enhanced mitochondria turnover by autophagy, and protected SH-SY5Y cells from rotenone-induced apoptosis [124]. EGCG promoted AMPK/mTOR/autophagy pathway and protected HEK293T cells against endoplasmic reticulum stress [125].

In non-canonical Beclin-1 independent pathway, autophagy is induced in Atg5/Atg7 dependent or independent ways. Resveratrol, genistein, and quercetin activate sirtuins and modulate autophagy either directly by promoting deacetylation of Atg5, Atg7, and Atg8, or indirectly by regulating FOXO3a transcription factor. Resveratrol activated AMPK/mTOR pathway and Sirt1, increased NAD^+^/NADH ratio, and promoted LC3 deacetylation and clearance of damaged mitochondria and Aβ by autophagy–lysosome system [126,127]. Resveratrol activated autophagy, degraded αSyn, and ameliorated the motor impairment and pathological changes in MPTP-induced mouse model of PD [128]. Resveratrol inhibited mTOR in competition with ATP and induced autophagy in MCF7 cells [129]. EGCG induced Sirt1, increased LC3-II, induced autophagy, and protected primary neuron cells against cytotoxicity of prion protein (106–126) [130].

Polyphenols also can prevent amyloidogenesis, inhibit the accumulation of Aβ and αSyn in mitochondria, and protect mitochondria from their cytotoxicity. Flavonoids (baicalein, kaempferol, catechin, EGCG, myricetin), theaflavins, oleocathal (a phenolic compound of extra-virgin olive oil), rosmarinic acid (an ester of caffeic acid and 3,4-dihydroxyphenyllactic acid), tannic acid, and curcumin inhibited formation and accumulation of the oligomers and fibrils [131,132]. Polyphenols (curcumin, EGCG, apigenin, baicalein, genistein morin, myricetin, rosmarinic acid, 2′,3′,4′-trihydroxyflavone), and black tea extract promoted clearance of amyloid aggregates of Aβ, wild and mutant αSyn (A30P, A53T) and tau-441 protein, and prevented the MPT [133,134,135,136,137].

Mitophagy is a beneficial process for cells by cleavage and exclusion of damaged mitochondria and accumulated amyloid deposition. Curcumin, resveratrol, and flavonoids (quercetin EGCG, genistein) most markedly activate the mTOR-regulated gene expression and the posttranscriptional modification of mitophagy-related factors, in addition to promoting the mitogenesis, mitochondrial dynamics, and mitophagy, suggesting that they might be the most promising neuroprotective phytochemicals at least in vitro.

## 4. Polyphenols Induce Cell Death in Cancer by Activation of Mitochondrial Apoptosis System

Most polyphenols exhibit both neuroprotective and anticancer activity in mitochondria. It remains elusive how they affect the mitochondrial apoptosis system in opposite directions. Cancer cells have an altered redox homeostasis and hyperpolarized mitochondria induced by the shift from mitochondrial respiration to aerobic glycolytic ATP synthesis, called the Warburg effect. The metabolic changes contribute to the compensation of the ATP synthesis in mitochondria and the enhancement of mitochondrial stabilization and resistance to apoptosis. Anticancer drugs acting on mitochondria are referred to “mitocans”, and now classified into (I) HK inhibitors; (II) mimickers of Bcl-2 homology-3 (BH3) domain; (III) thiol redox inhibitors; (IV) deregulators of VDAC/ANT complex; (V) ETC-targeted agents; (VI) lipophilic cations targeting the IMM; (VII) TCA cycle-targeting agents; and (VIII) mtDNA-targeted agents [138]. The anticancer function is also ascribed to deregulation of Ca^2+^ homeostasis, activation of cellular signal pathways to induce cell cycle arrest, modulation of transcription factors and expression of pro-survival genes, and induction of the MTP.

Natural polyphenols have been proposed as novel therapeutic agents for prevention and treatment of cancer in order to overcome multi-drug resistant of tumor cells. The antioxidant activity, intervention in cellular signal pathways, and activation of mitochondrial apoptosis system are proposed as the mechanisms behind the anticancer function [139,140,141]. Flavonoids (genistein, quercetin), curcumin, resveratrol, and astaxanthin directly or indirectly stimulate the mPTP formation and induce cell death in cancer cells. Quercetin and acacetin were most potent to induce apoptosis in human epidermoid carcinoma KB and KBv200 cells among tested flavonoids, and the hydroxyl group at position 3′ and the methoxy at 4′ in the B ring increased anticancer activities [142]. Curcumin bound to ANT, opened the mPTP, and induced apoptosis in WM-115 melanoma cells [143]. Apigenin found in chamomile flowers, thyme, and citrus fruits bound to ANT2, post-transcriptionally upregulated death receptor 5 and induced apoptosis by Apo2L/TRAIL (Apo2 ligand/tumor necrosis factor-related apoptosis-inducing ligand) [144].

Anticancer polyphenols inhibited or promoted the ETC activity depending on the concentrations. Resveratrol directly bound to complex I and III and the F-ATPase and inhibited the ETC activity [145,146,147]. Green tea-derived EGCG inhibited complex I and II and ATP synthase, and induced apoptosis in human malignant pleural mesothelioma cells, but not in normal mesoethelial cells [148]. Xanthohumol [a prenylflavonoid extracted from hops (*Humulus lupulus*)] and fisetin inhibited complex I [149,150], genistein complex III [151] and chrysin complex II and V [152], leading to the mPTP-mediated apoptosis in cancer cells.

Catechins (EGCG, galocatechin, gallocatechin gallate) inhibited anti-apoptotic Bcl-xL and Bcl-2 by binding with the gallate group [153]. AT-101, an acetic acid form of (-)-gossypol [2,2′-bis(8-formyl-1,6,7-yrihydroxy-5-isopropyl-3-methylnaphthalene)], isolated from cottonseed has potent cytotoxicity in cancer cells. AT-101 inhibited anti-apoptotic Bcl-2 and Bcl-xL, upregulated BH3 protein (PUMA, NOXA), downregulated X-linked inhibitor of apoptosis protein (XIAP), released apoptogenic Smac, and activated Akt/p53 pathway and SAPK/(c-Jun *N*-terminal kinase) pathway to induce apoptosis in cancer cells [154]. Astaxanthin and curcumin downregulated Bcl-2, Bcl-xL and IAP, upregulated Bax, and induced the mPTP opening and apoptosis in cancer cells [155,156].

HKs support the glycolysis highly required in cancer cells and protect cancer cells against apoptosis by binding to the cytosolic site of VDAC. Curcumin downregulated the expression and activity of HKs in human colorectal cancer HCT116 and HT29 cells [157]. Oroxylin A detached HK from mitochondria and inhibited glycolysis in human breast cancer MDA-MB-231 and MCF-7 cells [158]. Resveratrol suppressed glucose uptake, inhibited 6-phosphofruto-1-kinase, and decreased viability of MCF-7 cells [159]. Resveratrol also activated hypoxia-inducible factor 1α (HIF-1α), inhibited glucose uptake, and induced cell death in Lewis lung carcinoma HT-29 colon and T47D breast cancer cells [160].

Polyphenols perturb Ca^2+^ homeostasis in mitochondria and induce cell death. Curcumin increased Ca^2+^ influx into cells, depolarized ∆Ψm, released Cytc, and activated caspases in human hepatocellular carcinoma cells [161]. Honokiol (5,3′-dially-2,4′-dihydroxybiphenyl), eugenol (4-ally-2-methoxyphenol), gallic acid, and baicalein caused phospholipase C-dependent Ca^2+^ release from the endoplasmic reticulum (ER) and Ca^2+^ influx into cells through the transient receptor potential channel melastatin 8, or PKC-dependent channel in human glioblastoma and breast cancer cells [162,163,164]. Morusin (an extended 5,2′,4′-trihydroxyflavone) induced cell death by VDAC-mediated Ca^2+^ overload in epithelial ovarian cancer [165]. Resveratrol enhanced intracellular Ca^2+^ levels via low-threshold voltage-dependent T-type Ca^2+^ channels (Cav3.1, Cav3.2) in mesothelioma cell line [166].

Mitophagy has either tumor-suppressing or –promoting effects in cancer cells. In the early oncogenic process, autophagy suppresses tumor onset through maintaining genetic stability and modulating cellular ROS, whereas malignant transformation is associated with the defect of autophagy and induction of autophagy prevents tumor formation and development. Polyphenols induce autophagy-associated cell death in cancer cells by modulating autophagy-related proteins, transcription factors and signal pathways [167]. Rottlerin (a polyphenol isolated from fruits of *Mallotus philippomemsis*), genistein, quercetin, curcumin, and resveratrol activated signal pathways and induced autophagic cell death in cancer cells. Baicalein activated AMPK/ULK1, downregulated mTORC1 complex and induced cell death in human carcinomas [168]. EGCG suppressed LC3 expression, inhibited autophagy, and induced apoptosis in hepatoma cells [169]. Resveratrol increased fission proteins (Fis1, Deo1), ER stress, and MiR-326/pyruvate kinase M2, and induced cell death in human cervical (HeLa), colon (DLD1), breast (MCF-7), and liver (HepG2) cancer cell lines [170]. Polyphenols have been confirmed as potent anticancer compounds, and the cytotoxicity is mediated by the same mitochondrial apoptotic cascade and systems for dynamics and mitophagy, and signal pathways as for neuroprotection. The ambivalent function of polyphenols has been observed mainly in cellular models, but also in vivo in animal models and preclinical studies, as discussed below.

## 5. Synthesis of Mitochondria-Targeting Polyphenols

Development of “mitochondriotropic” compounds is an attractive perspective for neuroprotective and anticancer therapy. The matrix-negative voltage difference of about 180 mV promotes the selective accumulation of cationic compounds in mitochondria, and a 10-fold accumulation is expected by every-60 mV voltage difference. Compounds linked to a membrane-permeable cation triphenylphosphonium (TPP^+^) accumulate in negative-charged cell components, cytoplasm, and mitochondrial matrix, and various compounds are synthesized for therapeutic and diagnostic agents [171]. TPP-resveratrol derivatives increase uptake by 5–10 folds into cells across plasma membrane different in 30–60 mV voltage and by 100–500 folds into mitochondria different in 150–180 mV [172]. TPP group of the TPP conjugates binds to membrane at the hydrophobic site of the lipid/water interface and the hydrophobic tail penetrates into the membrane core. MitoQ_10_ based on coenzyme Q and SkQ1 [10-(6′-plastoquinyl)decyl TPP] derivatives based on plastoquinone were covalently linked to TPP by a 10-cabone alkyl chain and proposed as mitochondriotropic antioxidants for therapy of AD, PD, and age-related diseases [173]. TPP conjugates of quercetin, resveratrol, and caffeic acid were reported to improve mitochondrial activity [174]. TPP-caffeic acid derivatives prevented lipid peroxidation and inhibited the mPTP in cellular experiments [175], but they are cytotoxic and not applicable in vivo as neuroprotective compounds, unless the cytotoxicity can be minimized by such as modification with polyethylene glycol [176].

In PD, BDNF and GDNF have been shown to be one of the most promising neuroprotective compounds in clinical trials [177]. Since NTFs cannot pass the BBB, the synthesis of neuroprotective derivatives of BBB-permeable polyphenols, such as lipophilic flavonoids, alkaloids, courmarin, ginkgolide B, and terpenes, has been proposed to enhance in situ NTF synthesis in the brain and promote neuronal survival, but at present the effective compounds have not been developed [10,48]. On the other hand, derivatives of inhibitors of catabolic enzymes of neurotransmitters were synthesized based on the structure of polyphenols, which can enter into the brain depending on the degree of lipophilicity stereochemistry and interaction with efflux transporters, such as P-glycoprotein [178]. Pyrazolinemito derivatives of flavones, sesamol, and curcumin were synthesized to inhibit MAO, a major catalyzing enzyme of monoamines [179,180]. Ferulic acid was conjugated with memoquin, an inhibitor acetylcholinesterase (AChE) and β-secretase [181] and homoisoflavonoid Mannich-based derivatives modified Aβ aggregation and protected cells [182]. Genistein derivatives with carbon spacer-linked alkylbenzylamines inhibited AChE activity and prevented Aβ aggregation by the antioxidant and metal-chelating activities [183]. CNB-001 a cyclohexyl bisphenol derivative of curcumin increased bioavailability, suppressed neuroinflammation, induced Bcl-2, and showed neuroprotection in MPTP-treated mouse model of PD [184].

In cancer cells, mitochondria have higher transmembrane potential than in normal cells, which facilitates the uptake of cationic derivatives into cells and mitochondria. Resveratrol and quercetin were conjugated with a TPP group at the end of a four-carbon linker connected to the hydroxyl group at 3 in the C ring or 4′ in the B ring, which inhibited complex I and III and F_1_-F_0_ ATP synthase, increased superoxide production, and induced transient Ca^2+^ spike and cell death in tumor cells in vitro [185]. TPP derivatives of betulin (a triterpenid) were reported to promote cellular and intra-mitochondrial accumulation and enhance anti-proliferative activity in vinblastine resistant human breast cancer (MCF-7/Vib) cells [186]. TPP-furocoumarin [7*H*-furo(3,2-gichromen-7-one), psoralen] conjugate blocked the potassium channel Kv1.5 at the IMM and induced apoptosis in cancerous cells [187]. To promote neuroprotective potency, synthesis of highly BBB permeable compounds has been proposed as a novel strategy. Finding of specific transport system of polyphenols similar to that of other nutrients (amino acids, glucose, vitamins, and iron) has been tried, but at present the selective transport system has not been identified. Polyphenol metabolites, bioavailable phenolic sulfates derived by colonic metabolism of berries, were found to permeate the BBB and enhance bioactivity in vivo [188]. Novel neuroprotective compounds may be expected by synthesis of derivatives with BBB-permeable polyphenol metabolites with smaller mass.

## 6. Discussion

Dietary polyphenols, including flavonoids, curcumin, and resveratrol, show both beneficial and deleterious effects [189]. Antioxidant polyphenols function often as pro-oxidants, open the mPTP and promote cell death, and single electron oxidation of polyphenols causes the cytotoxicity [190]. The dualistic functions are induced by the redox activity, concentration, physiochemical property, and structures of particular polyphenol, and also by cellular conditions, such as redox context, pH, and oxygen concentration [191]. The concentrations of polyphenols affect the redox properties in biphasic ways, and are primarily associated with the ambivalent functions. Flavonoids (quercetin, myricetin, EGCG) are unstable and produce hydrogen peroxide depending on the concentrations and hydroxyl substituent patterns. The hydroxyl groups and double bonds in flavonoids allow interaction with proteins and lipids to define cellular redox state and activate redox signals. EGC and EGCG at the concentrations higher than 100 µM generate hydrogen peroxide and mediate DNA damage, but at 1–15 µM suppress them. The redox potency of polyphenol is associated with the inhibition of the MPT. Resveratrol at lower than 50 µM enhanced mitochondrial antioxidant function and prevented the mPTP opening through AMPA/Sirt1 pathway, but at higher than 50 µM induced the MPT and apoptosis through deregulation of Ca^2+^ homeostasis [192]. Ease-to-oxidation quercetin inhibited the mPTP formation, whereas kaempferol and catechin were less readily oxidizable and could not inhibit the MPT, even the similar structure and functional groups [193]. EGCG quinone produced by autoxidation and curcumin were reported to bind covalently to the thiol groups of cysteinyl residues in mitochondrial membrane protein and open the mPTP [194]. Hydrophobicity and the three-dimensional topographic structure are also associated with the cytotoxicity [195]. The studies using three-dimensional topographic structures have shown the correlation of steric and hydrophobic fields of polyphenols with apoptosis. Substituents with larger hydrophobicity at position 2 and less at 4, and those with more electronegativity at 4 and less at 3 increased the activity [196]. The position of hydroxyl groups, dipole moment magnitude, and shape of molecules correlate the antioxidant potency. Flavonoids inhibit matrix metalloproteinase (MMP) expression, depending on the hydroxyl group number in total of 3′,4′-*ortho*-dihydroxl in the B ring and 3-hydroxyl group in the C-ring. Anthocyanin (delphinidin) and flavonol (myricetin, quercetin, luteolin, apigen) inhibited the expression more markedly than other flavonoids. EGCG and GCG containing phenol hydroxyl groups at position 3′, 4′, and 5′ in the B ring inhibited the MMP activity more potent than ECG and CG containing the hydroxyl groups at position 3′ and 4′ [197].

Polyphenols interact with signaling molecules involved in cell survival and death. Figure 6 presents how polyphenols regulate mitogenesis by activation of cellular signal cascades. Polyphenols activate cell-survival signaling pathways including PKC, MAPKs (mitogen-activated protein kinase) and PI3K/Akt [198,199]. PKC is activated by phospholipids and Ca^2+^ and mediates normal cell function. EGCG and genistein increase PKCα and PKCε and protect neurons from apoptosis induced by neurotoxins. MAPK signal pathways mediate expression of pro-survival genes, including antioxidant enzymes, NTFs, and cytokines. MAPKs are classified into ERK, JNK, and p38. ERK1/2 phosphorylate CREB and upregulate Bcl-2 and Bcl-xL, and JNK regulates the transcription-dependent apoptotic signals, whereas p38 is implicated in cell death, cell cycle, senescence, and carcinogenesis. Catechins (EC, ECG, EGCG), genistein, and quercetin upregulate ERK/CREB pathway, and myricetin downregulates p38 and JNK to promote cell survival. PI3K/Akt pathway activates transcription factors and increases levels of NTFs and antioxidants to protect neuronal cells. EGCG inhibited phosphorylation of proteins and kinases (JNK, JUN, MEK1/2, ERK1/2), suppresses transcriptional activity of activator protein 1 and cell transformation, and induces G_0_ and G_1_ cell cycle arrest in bladder cancer T24 and 5637 cells and inhibited proliferation and migration [200,201].

In the brain, neurons interact with glia, and glia-derived NTFs, cytokines, ROS/RNS, and Ca^2+^ activate signal transduction and gene expression in neurons to maintain the function and structure [202]. In cancer, microenvironment consisting of extracellular matrix, stromal cells, and immune cells, regulates growth of tumor cells and is proposed as a target for anticancer therapy [203]. The effects of phytochemicals in mitochondria are also affected by crosstalk with other intracellular organelles and extracellular microenvironment. Polyphenols exhibit NTF-like function in the brain through intervention of transcription transduction pathways, either by activation of NTF receptors, or by direct way [17,204]. Resveratrol modulated signal pathways in tumor microenvironment and functioned as a pro-apoptotic compound in mouse tumor HL-I NB cells, but also as an anti-apoptotic one in normal cardiac cells [205,206]. Polyphenols can induce NTFs at concentrations lower than µM, much lower than those required for ROS scavenging, suggesting that polyphenols might protect cells in vivo mainly by activation of signal pathways [17]. These results suggest that the effects of polyphenols in mitochondria should be further investigated, taking these factors into consideration.

There is still great discrepancy between the results in cellular models and in clinical trials. The in vivo results are influenced by interactions with signal molecules, transport barriers, rapid metabolism, limited availability, and different bioactivity in purified form from the purified polyphenols. For understanding the biological effects of a particular polyphenol in human, the more refined experiments with more defined polyphenols should be designed in more selective subjects.

## Figures and Tables

**Figure 1 ijms-20-02451-f001:**
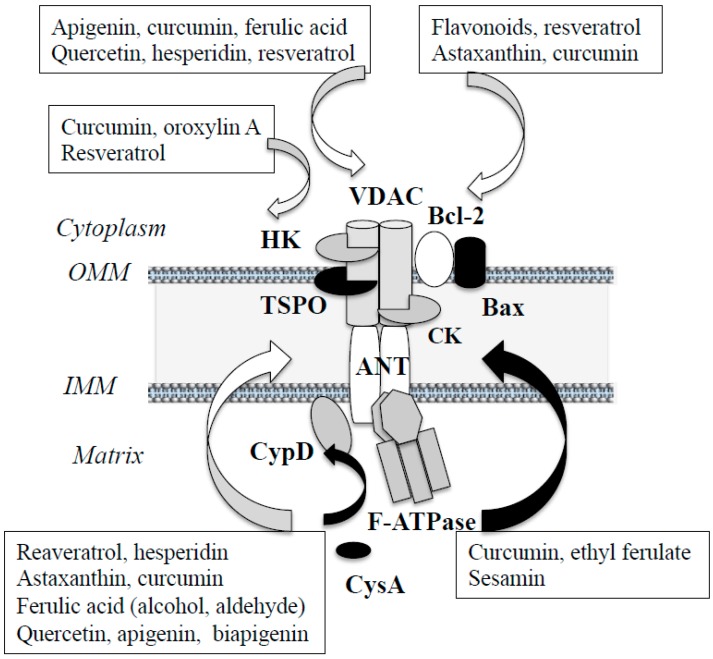
Schematic structure of the mitochondrial permeability transition pore (mPTP) and effects of phytochemicals on the mPTP components. Voltage-dependent anion channel (VDAC) and adenine nucleotide translocator (ANT) are considered as major mPTP components. The outer membrane transporter protein (18 kDa) (TSPO) interacts with the mPTP at the outer mitochondrial membrane (OMM), hexokinase (HK) from cytoplasm, creatine kinase (CK) at the intermembrane space, and cyclophilin D (CypD) interacts with ANT from the mitochondrial matrix. But it has not been established whether they are absolutely required as the pore constituents. F_1_-F_0_ ATP synthase (F-ATPase) binds to CypD and the c subunit ring of the F_0_ domain is included into the pore-forming unit of the mPTP. CypD binds to ANT at the IMM and induces the membrane permeability transition (MPT) at the IMM, which cyclosporine A (CysA) inhibits. Anti-apoptotic Bcl-2 proteins located at the OMM inhibit the mPTP opening. Phytochemicals interact with the mPTP components either in a suppressive or promoting way, as shown by white or black arrows. White and black arrows indicate promoting and inhibiting effects, respectively.

**Figure 2 ijms-20-02451-f002:**
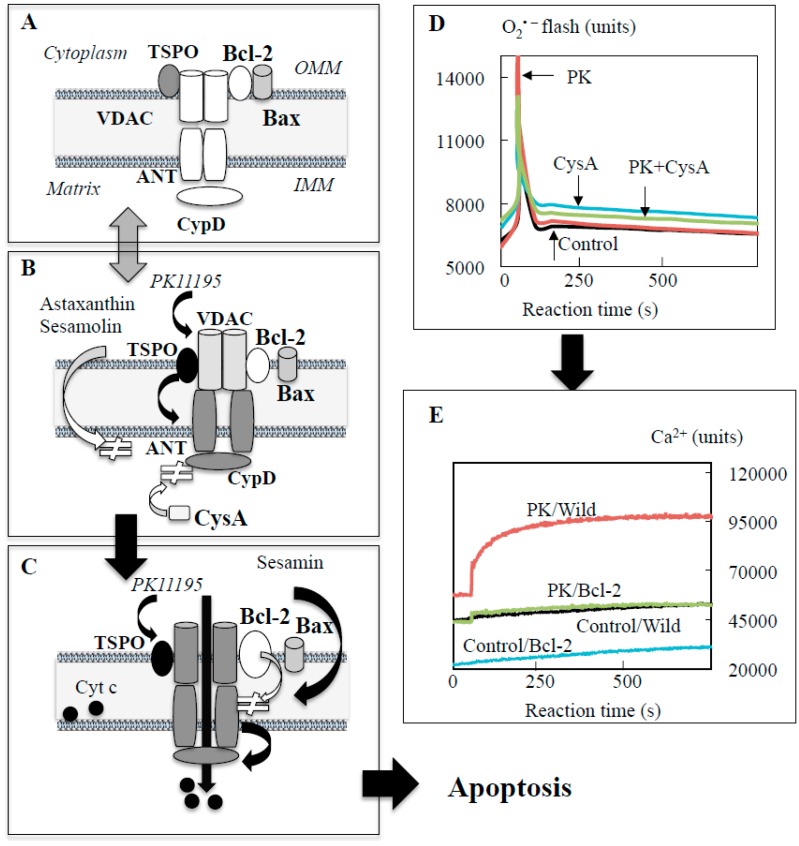
Step-wise formation of the mitochondrial permeability transition pore (mPTP) induced by PK11195 and regulation of cyclosporine A (CysA) and Bcl-2. (**A**) The mPTP at the resting state. (**B**) PK11195 opens transitionally reversible pore at the IMM, induces superoxide flash and ΔΨm decline, and allows the influx of water and molecules with less than 980 Da. (**C**) Excess and prolonged stimuli open the mPTP, efflux Ca^2+^, increase the permeability of molecules up to 1500 Da, including cytochrome c (Cytc) and other caspase activators, and initiates apoptosis cascade. (**D**) Superoxide was visualized by chemiluminescent method. Immediately after PK11195 addition, superoxide burst was detected and CysA pretreatment suppressed the superoxide flash. PK, SH-SY5Y cells treated with PK11195; CysA, with CysA; PK + CysA, cells pretreated with CysA, then with PK11195. **E**. Ca^2+^ in the cytoplasm detected by fluorescent method increased gradually after PK11195 addition and reached to plateaus, which Bcl-2 overexpression could prevent. Control/Wild and Control/Bcl-2, wild and Bcl-2 overexpressed SH-SY5Y cells as control; PK/Wild and PK/Bcl-2. wild and Bcl-2 overexpressed cells treated with PK11195. White and black arrows indicate promoting and inhibiting effects, respectively.

**Figure 3 ijms-20-02451-f003:**
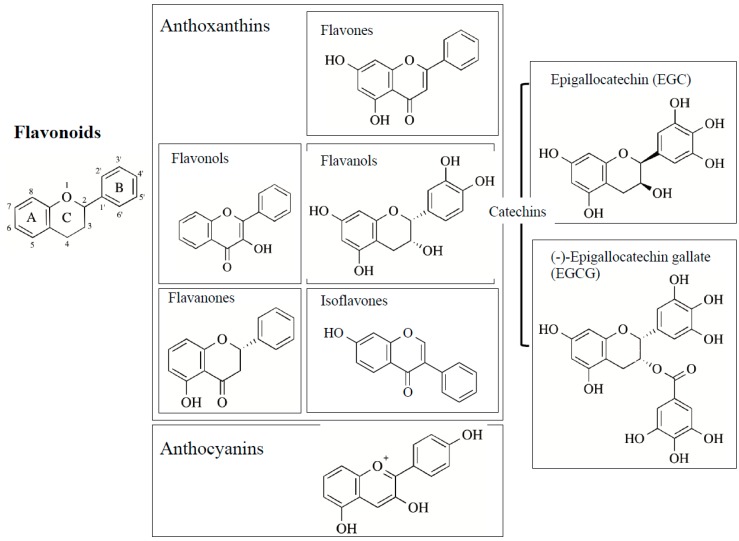
Classification and structures of major flavonoi s. Flavonoids are classified into anthoxanthins (flavonols, flavanones, flavones, flavanols and isoflavones) and anthocyanins, depending on the number and position of hydroxyl groups. Chemical structures of epicatechin-3-gallate (EGC) and epigallocatechin-3-gallate (EGCG) are also shown. The chemical structures of individual flavonoids are shown in Table 1.

**Figure 4 ijms-20-02451-f004:**
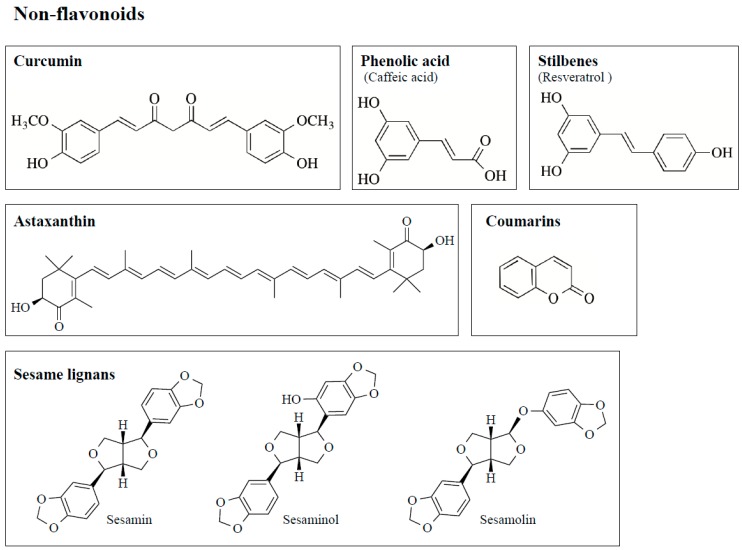
Structures and classification of major non-flavonoid polyphenols. Chemical structure of curcumin, phenolic acids, stilbenes (resveratrol), astaxanthin, coumarin, and sesame lignans are shown. Phenolic acids are divided into two groups; benzoic acid (gallic, protocatechulic acid) and cinnamic acid derivatives (caffeic, ferulic, *p*-coumaric acid) and the chemical structures are shown in Table 2.

**Figure 5 ijms-20-02451-f005:**
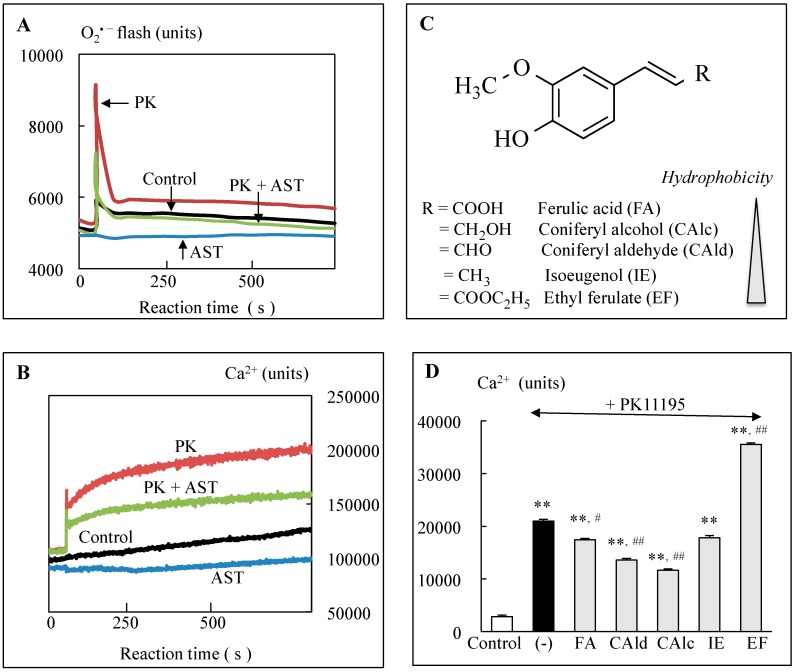
Phytochemicals inhibited or promoted the mitochondrial permeability (mPTP) opening. (**A**,**B**) Astaxanthin (AST) inhibited the PK11195-induced mPTP, which was detected as superoxide flash (**A**) and Ca^2+^ efflux (**B**). PK, AST and PK + AST: SH-SY5Y cells treated with PK11195, AST, and cells pretreated with AST then PK11195. (**C**) Chemical structures of ferulic acid derivatives, which shared common 3-methoxy and 4-hydroxy groups on the benzene ring for antioxidant potency. The side chains are associated with the amphipathic properties. (**D**) The effects of ferulic acid derivatives on PK11195-induced Ca^2+^ efflux. **, significantly different from control, *p* < 0.01. #, and ##, significantly different between PK11195-treated cells without and with pretreatment of ferulic acid derivatives, < 0.05 and 0.01, respectively.

**Figure 6 ijms-20-02451-f006:**
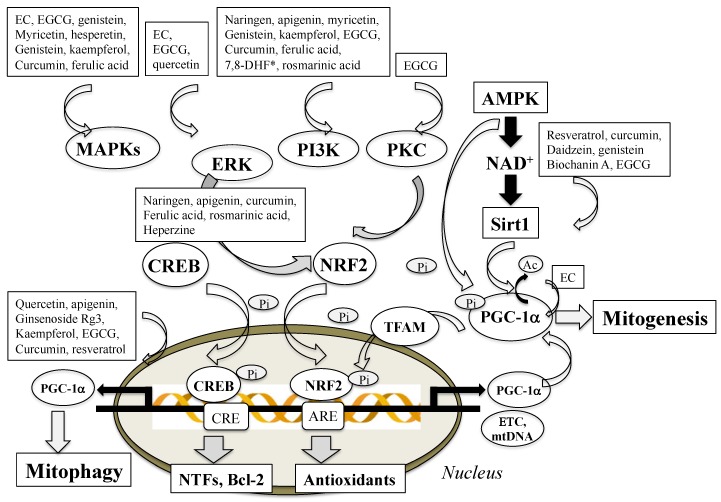
Effects of polyphenols on cellular signal transduction to activate mitogenesis. Polyphenols directly activate peroxisome proliferation-activated receptor γ coactivator 1α (PGC-1α) by deacetylation or phosphorylation through AMP-activated protein kinas (AMPK)-situin-1 (Sirt1) pathway. Polyphenols also enhance PGC-1α expression by mitochondrial transcription factor A (TFAM)/CREB signal pathway and protein kinases/Nrf (nuclear respiratory factor-1)/ARE (antioxidant responsive element) pathway. In addition, polyphenols activate other protein kinases and transcription factors and increase the expression of genes regulating mitochondrial function and homeostasis. * 7,8-dihydroxyflavone. White and black arrows indicate promoting and inhibiting effects, respectively.

**Table 1 ijms-20-02451-t001:** Chemical structure of cited flavonoids. * O-rutinoside.

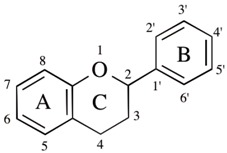								
**Flavonols**								
Name	3	5	7	2′		3′	4′	5′
Datiscetin	H	OH	OH	OH		H	H	H
Fisetin	H	H	OH	H		OH	OH	H
Galangin	H	OH	OH	H		H	H	H
Kaempferol	H	OH	OH	H		H	OH	H
Morin	H	OH	OH	OH		H	OH	H
Myricetin	H	OH	OH	H		OH	OH	OH
Quercetin	H	OH	OH	H		OH	OH	H
Rutin	ORu *	OH	OH	OH		OH	OH	H
**Flavanols**								
Name	3				5′			
Catechin	H				H			
EGC	H				OH			
EGC	H				OH			
EGCG	Gallate				OH			
**Flavones**								
Name	6		8		3′		4′	
Acacetin	H		H		H		OCH_3_	
Apigenin	H		H		H		OH	
Baicalein	OH		H		H		H	
Chrysin	H		H		H		H	
Eupafolin	OCH_3_		H		OH		OH	
Hispidulin	OCH_3_		H		H		OH	
Luteolin	H		H		OH		OH	
Oroxylin A	OCH_3_		H		H		H	
Wogonin	H		OCH_3_		H		H	
**Isoflavones**								
Name	5		7		3′		4′	
Biochanin A	OH		OH		H		OCH_3_	
Daidzein	H		OH		OH		H	
Formononetin	H		H		OCH_3_		H	
Genistein	OH		H		OH		H	
**Flavanones**								
Name	7			3′		4′		
Hesperidin	ORu			OH		OCH_3_		
Hesperetin	OH			OH		H		
Liquiritigen	OH			H		OH		
Naringenin	OH			H		OH		
**Anthocyanins**								
Name	7			3′		5′		
Cyanidin	H			OH		H		
Delphidin	H			OH		OH		
Malvidin	OH			OCH_3_		OCH_3_		

**Table 2 ijms-20-02451-t002:** Structures of phenolic acids.

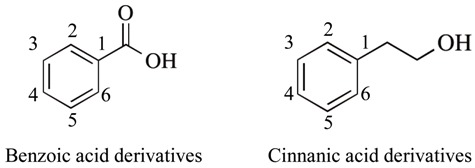			
Name	3	4	5
**Benzoic acid**			
Gallic acid	OH	OH	OH
Protocatechuic acid	OH	OH	H
Vanillic acid	OCH_3_	OH	H
**Cinnamic acid**			
Caffeic acid	OH	OH	H
Ferulic acid	OCH_3_	OH	H
*p*-Coumaric acid	H	OH	H

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
