# Peer review of "Mitochondria in Neuroprotection by Phytochemicals: Bioactive Polyphenols Modulate Mitochondrial Apoptosis System, Function and Structure"

_ijms, 2019, doi:10.3390/ijms20102451_

Reviewer 1 Report

Oxidative stress, deficit of neurotrophic factors and other multiple factors impair mitochondrial function. This review article focused on the phytochemicals that can scavenge reactive oxygen and nitrogen species to introduce the potentials. It is prepared in a good way. But, I like to give the minor suggestions.

1.      Abbreviations did not include the all used in this report. Please check it again.

2.      Dual effects of phytochemicals on apoptosis were not conducted in clear. What is the key-point in production of various effects?

3.      The potential one from phytochemicals could be suggested in addition.

Author Response

Answers to the comments of Referee #1

According to comments from you and other referees, I revised my paper and corrected parts were shown in red.  The part of the Title was changed and mistakes were corrected through the text.  Minor points in Figs. 1 and 5C were also corrected.  As a referee advised, “phytochemical” is a too broad term, and it was changed into “polyphenol”, a more specified term.  In Session and Discussion, the text was revised to make the point clearer.  New references were added, according to referee’s comments.

Thank you for your advice and comments, and I tried hard to improve my paper.

1.        I checked abbreviations and added ones used more than twice in the text to the list.

2.        In Discussion, I correct the sentence t emphasize that the concentration-dependent redox activity may play a key role in the ambivalent properties of phytochemicals.

3.        It is a very difficult question.  Maybe, quercetin, EGCG, curcumin and resveratrol are potential protective compounds at least in vitro.  I added one sentence in Discussion.

Reviewer 2 Report

The manuscript is well constructed and show the various polyphenols activity in relationship with the mitochondrial protection and neuroprotection. There are following minor comments need to be corrected:

Page 1; Line 18: Phytochemical is a very broad term, however author have described most of the activity from polyphenols or flavonoids. Need to be corrected throughout the manuscript.

Page 2; Graphical Summary: Should be corrected

Page 3; Line 59: Need reference here.

Page 4; Line 140: Should be "Polymorphisms in mtDNA"

Page 6; Figure 1: All the abbreviation in the picture should be defined here again to make the diagram self explanatory.

Page 7; Line 212: Heading should be corrected with Polyphenols.

Page 8; Line 224: Heading should be corrected with Polyphenols.

Page 11: Line 282: Delete repeated word Curcumin with "it"

Page 11: Line 283: Need reference.

Page 15; Fig 6: All the abbreviation in the picture should be defined here again to make the diagram self explanatory.

Page 19; Line 582: Usually sentence can't be started with abbreviation.

Author Response

Answers to the comments of referee #2

According to comments from you and other referees, I revised my paper and corrected parts were shown in red.  The part of the Title was changed and mistakes were corrected through the text.  Minor points in Figs. 1 and 5C were also corrected.  In Session and Discussion, the text was revised to make the point clearer.  New references were added, according to referee’s comments. 

Thank you for your advice and comments, and I tried hard to improve my paper.

Page. 1.  As you commented “phytochemical” is rather too broad termed.  I checked the terms through out the text, and changed “phytochemicals” with “polyphenols”.

Page 2.  Graphic summary was corrected.

Page 3, line 60. A new reference was added.

Page 4, line 146 (now 149).  I corrected to “Polymorphisms in mtDNA”.

Page 6.  In legends for Figs. 1, 2 and 6, the abbreviations were defined.

Page 8, 13, 15, 16, 17, 18   Phytochemicals in the title and headings were replaced with polyphenols.

Page 11, line 290 (now 203).  Curcumin was corrected as “it”.

Page 11, line 294 (now 304).  A reference was cited

Page 18, line 586 (now 608).  “EGCG” was changed to “A green tea-derived catechins (EGCG,   ”

Reviewer 3 Report

General Remarks

1.   The review is very rich in details. Overload of facts and citations with minimal narration makes it difficult to follow. The article encompasses immense range of phytochemicals derived from various sources. It also describes in detail the mitochondrial mechanisms of functioning and apoptosis regulation in health and disease of the nervous system, mostly PD and AD, as well as in animal models and in vitro studies. It discusses the subject from different angles and multiple target points. The text is complicated although prepared meticulously. There are figures that facilitate summary of knowledge.

2.   Language needs polishing. Grammar needs corrections.

Specific Remarks

3.   The title: Mitochondria in neuroprotection by phytochemicals: Bioactive polyphenols regulate mitochondrial apoptosis system, function and structure. The word regulate indicate some permanent, key function in mechanism. Could be better to use “influences”, “affects” or similar.

Abstract:

4.   Line 10 “loss of distinct type of neurons characterizes disease-distinct pathological”. Change to ‘disease-specific’.

5.   Line 16: “Multi-functional phytochemicals are proposed as one of most promising mitochondria-targeting medicine to prevent the dysfunction and affect the function and structure, in order to maintain neuronal function, survival and differentiation”. 4x function in one sentence. Please rephrase. Dividing one long sentence to more shorter ones could benefit the reading flow.

6.   Line 21: “regulate mitochondrial apoptosis  system  either  in  preventing  and or promoting  way”.

7.   A closing sentence would be good at the end of abstract. Or rephrasing the existing one.

Introduction:

8.   Line 46: Mitochondria  are  originated originate from  bacterial  endosymbions

9.   Line 87: have been proven not have been proved.

10.  Line 89: “Epidemiological studies present that diet and bioactive food factors [to some extent] can ameliorate decline in..”

11.  Line 91: “Antioxidant  activity  has  been  considered  as  the  major  neuroprotective  function  of  phytochemicals,  but  the [their] concentrations of phytochemicals are [is] too low to scavenge free radicals in the brain [13].”

12.  Line 98: “Phytochemicals  ambivalently  suppress  or  activate mitochondrial  death  system”.

Paragraph 2:

13.  Line 123: “In  Parkinsonism  patients  induced  by  1-methyl-4-  phenyl-1,2,3,6-tetrahydropyridine (MPTP),..” Please rephrase this sentence. MPTP caused parkinsonian syndrome in humans but not Parkinson’s disease. There was degeneration of dopaminergic neurons and behavioural deficits but Parkinson’s disease is usually called an idiopathic form of the disease. Parkinsonism or parkinsonian syndrome can be induced by toxins or head injury – anything that causes lesion in substantia nigra.

14.  Line 135: “mtDNA encodes 13 subunits of complex I (7 of the 49 protein subunits), III, IV and V.” This sentence is not clear. Please rephrase.

15.  Fig.1 and fig 6 capital vs small letters. Please make names uniform. Resveratrol and other names are sometimes written with big or with small letters. Same in the text line 282 Curcumin. Use commas in constant way on the figures.

Paragraph 3:

16.  Line 217: “Mitochondria-targeting  phytochemicals  have  been  devised  to  increase  the  selective  accumulation  and  protective  potency  in  mitochondria” Selective accumulation of what?

17.  Line 218: Cellular  mechanisms  behind neuroprotection by polyphenols have been intensively investigated in animal and cellular models  of  AD,  PD,  Huntington’s  disease  (HD),  amyotrophic  lateral  sclerosis  (ALS)  and  multiple sclerosis (MS),…” authors discussed mitochondrial pathology of mostly AD and PD earlier.

18.  Authors declare that several flavonoids can cross BBB. Do all described below molecules are able to do so? Please clarify and specify in the text. It is also valid if studies were performed in vitro or in vivo in context of affecting brain, as you indicated in several paragraphs, for example in resveratrol.

19.  Neuroprotective action is opposite to anti-cancer. It would be interesting to describe the difference. Authors indicate amphipathic properties of phytochemicals as the underlying feature of dualistic anti-apoptotic vs pro-apoptotic action. There is either opening or closing of mPTP and some substances do both. More impact on this dualistic issue would be advised and clearer division between pro- and anti-apoptotic substances.

20.  Line 351 “The  step-wise  pore  formation  at” is the continuation of the previous paragraph (?). The section indentation is unnecessary.

21.  Line 360, “in this model” In the same model? - similar as above, the section indentation is unnecessary.

22.  Line 468: “   Allicin (diallyl thiosulfinate) the main biological compounds derived from gall” Derived from gall or garlic?

23.  Line 516: “Loganin [7-hydroxy-6-desoxyvenalin, a compound derived from fruits of cornus (Cornus officinalis)]”. English vocabulary does not recognise word “cornus”. Its name is dogwood.

24.  Line 632: “Neuroprotective  phytochemical  derivatives  have  been  synthesized  by  strategy  not  targeting  mitochondria: increased permeability through the BBB and in situ synthesis of NTFs and Bcl-2 in the  brain.” I don’t understand this sentence.

25.  Line 678: “but interaction  between  “inversed”  mitochondria  and  the  “host”  cells  determines  the  neuroprotective “. I don’t understand this sentence.

26.  There is a vast amount of abbreviations in the text, even if symbols are found only once in the text. Some can be eliminated such as prion protein (PrP), etc. to facilitate reading.

27.  There is overload of facts and almost no summarizing narration. Just like encyclopedia.  Please write more introduction and conclusions for each paragraph. There is a story in what you are describing. Make it more readable.

Author Response

Answers to the comments of Referee #3

According to comments from you and other referees, I revised my paper and corrected parts were shown in red.  The part of the Title was changed and mistakes were corrected through the text.  Minor points in Figs. 1 and 5C were also corrected.  In Session and Discussion, the text was revised to make the point clearer.  New references were added, according to referee’s comments. 

Thank you for your advice and comments, and I tried hard to improve my paper.

1.        I failed to come out from a maze of multiple, diverse and dualistic phytochemicals. But, I tried to make my paper more compact and simpler, according to your advice.

2.        I tried to improve my paper and correct grammatical mistakes.

3.        I modified the title by changing “regulate” to modulate”.

4.        Thank you for your comment.  I changed to “specific”.

5.        I changed this sentence totally.

6.        I corrected, “and” to “or”.

7.        I added a sentence at the end of Introduction.

8.        I corrected “are originated” to “originate”.

9.        Thank you for your comment.  I changed “proved” to “proven”.

10.    I added ‘to some extent”, as you commented.

11.    I corrected, as you pointed out.

12.    I deleted “ambivalently”.

13.    I revised the sentence according to your advice.

14.    I revised this sentence.

15.    I tried to correct the mistakes.

16.    I added “of bioactive compounds”.

17.    I corrected this sentence and deleted the parts of other disease.

18.    The permeability of polyphenols across the BBB was discussed more in detailed in revised version in the Session 5.  Flavonoids permeate the BBB in vivo and in vitro were listed in page 11, line 288, and two new references were cited.

19.    The most interesting point in this paper is that the same polyphenol opens and closes the mPTP and functions as a neuroprotective and anticancer compound.  According to the comments from you and another referee, I revised Discussion to discuss in more details.

20.    I corrected the text.

21.    I corrected the part.

22.    I corrected to “garlic”. 

23.    I changed the part of the sentence.

24.    I totally rewrote the sentences in page 20, lines 675-682 (now 668-672).  Added references on the BBB-permeable flavonoids.

25.    I revised these sentences (now page 22, line 753).

26.    I deleted unnecessary abbreviation through the text.

27.    I included short sentences at the beginning and end of the sessions as the introduction and conclusion.